# Comparison of traditional systemic analgesic, single shot or continuous fascia iliaca compartment block for pain management in patients with hip or proximal femoral fractures: A protocol for systematic review and network meta-analysis

**Jia-Xi Tang**[1⊙], **Ling Wang**[2⊙], **Shaojin Bu**[3⊙], **Wallisa Roberts**[4⊙], **Narcis Ungureanu**[5], **Ansar Mahmood**[6], **Fang Gao**[5,7], **Raja V Lakshmanan**[5], **Tonny Veenith**[5,7*], **Rajneesh Sachdeva**[5,7*]

1 Department of Anesthesiology, Chongqing Key Laboratory of Translational Research for Cancer Metastasis and Individualized Treatment, Chongqing University Cancer Hospital, Chongqing, China, 2 Department of Phase I Clinical Trial Ward, Chongqing University Cancer Hospital, Chongqing, China, 3 Department of Anesthesiology, Fengdu People's Hospital, Chongqing, China, 4 Department of Emergency medicine, University Hospital of Coventry & Warwickshire, United Kingdom, 5 Department of Anesthesiology, University Hospitals of Birmingham NHS Foundation Trust, Birmingham, United Kingdom, 6 Trauma and Orthopaedics Division, University Hospitals Birmingham NHS Foundation Trust, Birmingham, United Kingdom, 7 Department of Inflammation and Ageing, University of Birmingham, Birmingham, United Kingdom

⊙ These authors contributed equally to this work, They are joint first authors.
* tveenith@gmail.com (TV); R.sachdeva@bham.ac.uk (RS)

## Abstract

### Introduction

Pain management for hip and proximal femoral fractures includes oral and parenteral opioids and various regional anesthesia techniques. Fascia iliaca compartment blocks (FICB) are commonly used for these patients. At present, a unified view of the analgesic effect of FICB has not been reached. In addition, the comparison between single shot FICB and continuous FICB has not elicited clear evidence-based results. We will compare the efficacy and safety of systemic analgesics, single shot or continuous FICB in the pain management, complication prevention and satisfaction, in our systematic review and network meta-analysis.

### Methods

China National Knowledge Infrastructure, Chinese Biomedical Literatures database, PubMed, the Cochrane Central Register of Controlled Trials, Physiotherapy Evidence Database, EMBASE, and Web of Science will be searched until June 2023. Two authors will independently screen the studies for eligibility and perform data extraction. The Cochrane risk of bias tool (RoB 2) will be used to assess the quality of evidence. We will

**Data availability statement:** No datasets were generated or analysed during the current study. All relevant data from this study will be made available upon study completion. All data from the study will be presented in subsequent manuscript and/or Supporting information files.

**Funding:** This research be funded by Key Laboratory Open Fund Project of Chongqing University Cancer Hospital grant to JXT (cquch-kfjj006, https://www.cqch.cn/) and Chongqing Science and Technology Bureau joint Health Bureau grant to SJB (2023MSXM033, http://kjj.cq.gov.cn/). The funders had no role in study design, data collection and analysis, decision to publish, or preparation of the manuscript.

**Competing interests:** The authors have declared that no competing interests exist.

use the GRADE approach to assess the certainty of the evidence across studies included in this review. All the statistical analyses will be conducted using Rev Man 5.3, WinBUGS 1.4.3, and Stata 13.

## Ethics and dissemination

Our review involves a secondary analysis of existing published studies, therefore there is no need for formal research ethics approval. We will disseminate our findings through publication in a peer-reviewed journal,

## Protocol registration

PROSPERO, CRD42023425282

## Introduction

Hip fractures or proximal femoral fractures are one of the most common and painful injuries in all ages [1,2]. With the global trend of population ageing, the number of hip fractures is expected to significantly increase [3–5]. According to the literature above, by 2033, annually around 100,000 UK residents will require surgery for femoral neck fractures. By 2050, this figure is projected to reach 250,000 in Germany for osteoporosis-related hip fractures. Globally, the estimated number of hip fracture sufferers is anticipated to soar to 6.26 million by 2050. Guidelines from the Association of Anesthetists of Great Britain and Ireland indicate that there is a high incidence of morbidity and mortality in patients with hip fractures, particularly in older patients [6]. To reduce the incidence of complications from hip fractures, the American Academy of Orthopedic Surgeons (AAOS) clinical practice guidelines recommend early surgical treatment for these patients [7]. Inadequate pain control during perioperative phases has been linked to diminished patient satisfaction, which in turn can hinder rehabilitation progress and even elevate the risk of developing delirium [8].

In summary, the current clinical analgesia methods for hip fractures include systemic analgesia, epidural analgesia and regional nerve blocks [9–14]. Systemic analgesics, including opioids and non-steroidal anti-inflammatory drugs (NSAIDs), provide relief by either inhibiting pain signaling in the central nervous system (opioids) or reducing inflammation (NSAIDs). However, these treatments alone may not offer sufficient pain relief, and their use in high doses often leads to adverse side effects [15]. Fascia iliaca nerve block (FICB) is a regional anesthetic technique that targets the femoral, lumbar plexus, and obturator nerves. By injecting a local anesthetic into the fascia iliaca space, FICB provides effective pain relief, reduces opioid use, and has fewer side effects. It is a valuable adjunct to systemic analgesics in the management of hip and proximal femoral fractures. FICB is currently the most commonly performed nerve block for these patients. The block is easy, safe and reliable [16].

At present, many systematic reviews have evaluated the analgesic effect of FICB, but a unified view on their universal use has not been reached. Slade et.al [17], Makkar et.al [18] and Fadhlillah et.al [19] reported that FICB can effectively reduce pain after hip fracture and reduce opioid consumption. However, studies by Salottolo et.al [20] and Hayashi et.al [21] show that FICB does not reduce the need for opioids in patients with hip fractures. The studies conducted by Schulte et al [22] and Pasquier et al [23] indicate that FICB is not superior to the control group in terms of postoperative pain scores. Moreover, FICB could induce quadriceps weakness, which could potentially contribute to an elevated risk of falling in the postoperative period. In addition, the systematic review of comparison between single shot FICB

and continuous FICB has not elicited clear evidence-based results and recommendations. Also, there is a scarcity of studies directly comparing single shot FICB and continuous FICB.

Our hypothesis is that compared with single shot FICB or systemic analgesia, continuous FICB may prolong analgesic effects and reduce opioid use, but it may simultaneously prolong lower limb weakness, thereby increasing the risk of postoperative falls. To verify the hypothesis, we will utilize systematic review and network meta-analysis to compare the efficacy and safety of systemic analgesia, single-shot FICB, and continuous FICB for acute pain management during hospitalization in hip or proximal femur fractures. By comparing these methods, our study aims to provide evidence that helps clinicians optimize multimodal analgesia, balancing efficacy, safety, and resource considerations to meet the needs of diverse patient populations.

## Materials and methods

The systematic review and network meta-analysis protocol have been submitted to the International Prospective Register of Systematic Reviews (PROSPERO) under the registration number CRD42023425282. We will conduct a systematic review and network meta-analysis in adherence to the guidelines provided by the Preferred Reporting Items for Systematic Reviews and Meta-Analysis Protocols (PRISMA-P) 2015: elaboration and explanation [24] (S1 Table). For this study, we will follow the traditional pairwise and network meta-analysis methods as outlined in the Cochrane Handbook for Systematic Reviews of Interventions [25] and present all findings following the preferred reporting items for systematic reviews and meta-analysis for network meta-analysis (PRISMA-NMA) [26].

### Identification and selection of studies

**Data source.** We will identify relevant studies by an electronic search consisting of seven databases: China National Knowledge Infrastructure (CNKI, 1999 to June 2023) and Chinese Biomedical Literatures database (CBM, 1978 to June 2023), PubMed (1966 to June 2023), the Cochrane Central Register of Controlled Trials (CENTRAL, 1996 to June 2023), Physiotherapy Evidence Database (PEDro, 1999 to June 2023), EMBASE (1947 to June 2023) and Web of Science (1900 to June 2023). The electronic database search will be supplemented by a manual search of the reference lists of included articles and topic related reviews and electronically retrieved Clinicaltrial.gov.

**Search strategy and study selection.** The keywords and search strategy include: (hip fractures OR (Femoral Fractures NOT Femoral Fractures, Distal)) AND (Fascia iliaca block OR systemic analgesia) AND random. Retrieval strategies will be adjusted accordingly according to different databases. The full text of the articles is limited to English and Chinese. The search strategy of the PubMed database is shown in S2 Table.

After literature search, we will use EndNote to remove duplicate records, and then import the remaining studies into Rayyan for independent screening of titles and abstracts by two authors. Afterward, the full text of the selected studies will be downloaded for further evaluation. For any discrepancies during the screening process, the two authors will first resolve the issues through discussion, and if necessary, a third author will be involved for arbitration. The screening process is shown in Fig 1.

In our study, literature inclusion criteria were as follows: (i) study design: Only randomized controlled trials (RCTs) will be included. Abstracts will be considered if they provide sufficient data, defined as detailed information about the patient population, intervention (e.g., fascia iliaca nerve block), and at least one primary or secondary outcome related to analgesic efficacy or safety. Abstracts lacking this essential information, such as an absence of intervention description or

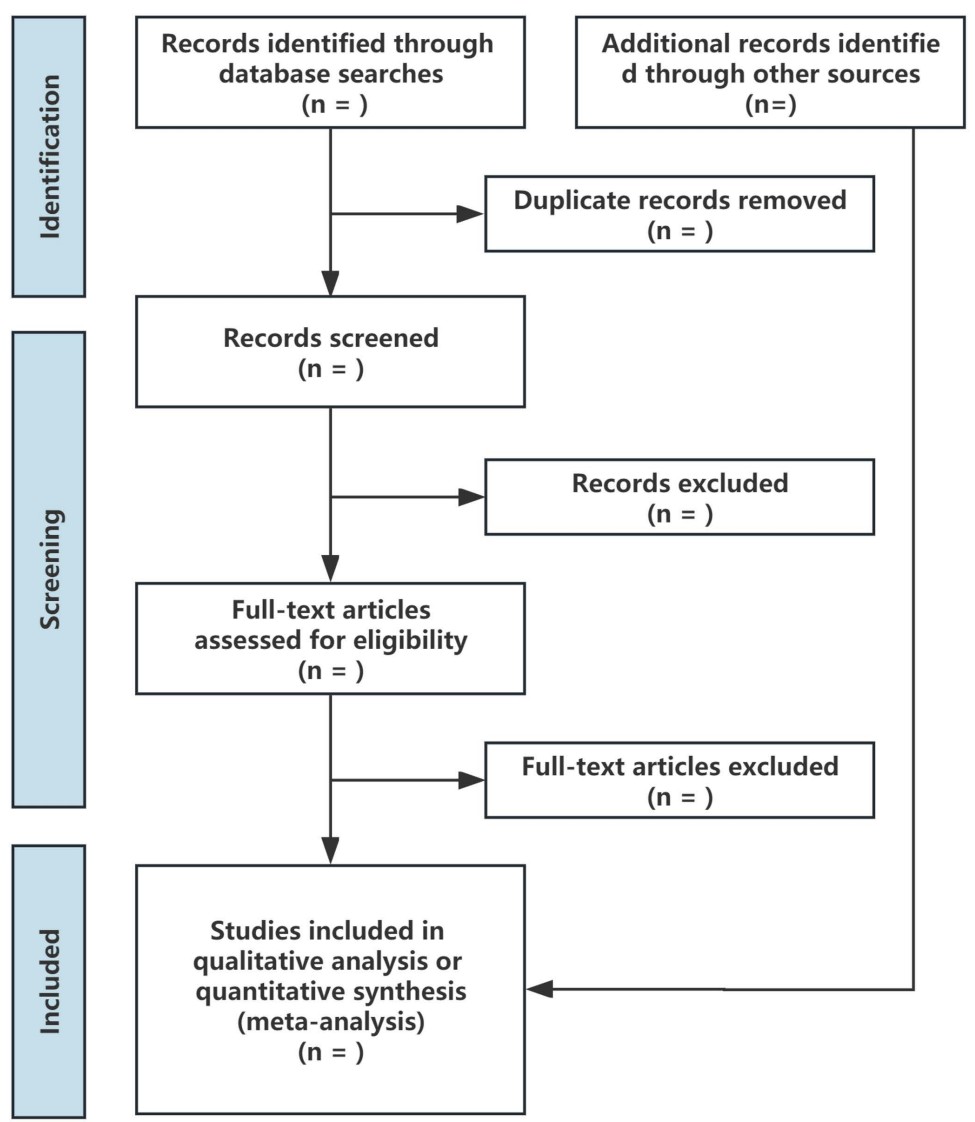

**Fig 1. Flow chart of studies search and selection.**

reported outcomes, will be excluded. (ii) participants: all patients with unilateral hip or proximal femur fractures, whether or not they underwent surgery. (iii) intervention: all analgesia regimens, including fascia iliac compartment block (single-injection or continuous infusion) with or without systemic analgesia and systemic analgesia alone. We will construct the evidence network based on the targeted regimens' effectiveness in analgesic (Fig 2). (iv) outcome measures: The primary outcome will be pain scores and the dosage of opioid. Secondary outcomes will include success rate, occurrence of adverse events (ARs), complications, patient satisfaction, length of hospital stay, duration of lower limb weakness. (v) language: there is no language restriction.

Studies that meet one of the following criteria will be excluded: (i) replicated studies; (ii) Low-quality studies with Jadad scores less than 2 [27]; (iii) non-original types of research, such as reviews, editorials or comments; (iv) research that fails to extract essential information; and (v) Studies that compare FICB with other nerve blocks techniques, or studies that compare other nerve blocks techniques with systemic analgesia.

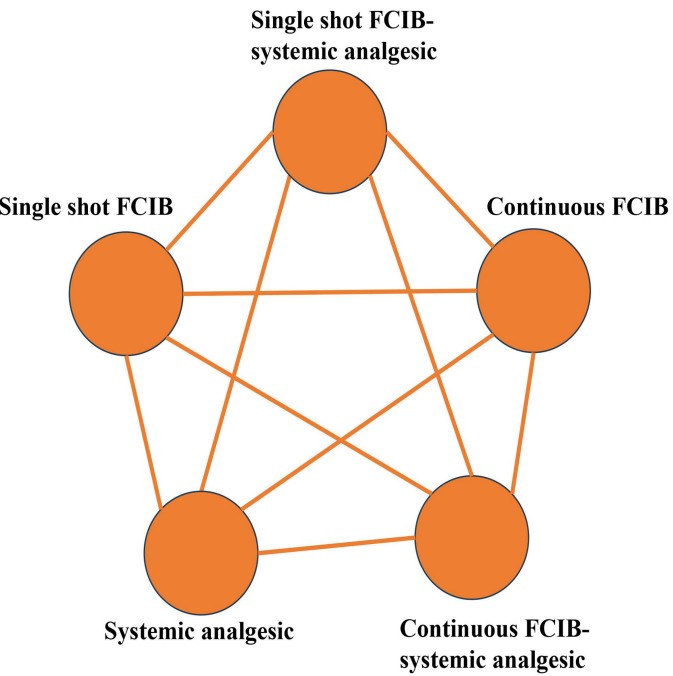

**Fig 2. Possible evidence network of all possible Pain management regimens in terms of Analgesic efficacy.** FICB, Fascia iliaca compartment block.

## Assessment of characteristics of studies

**Data extraction.** A standardized data extraction form will be used to ensure consistency in the collection of outcome data. The following data will be extracted: first author, year of publication, country, sample size, participant information, details of the intervention and random hidden allocation scheme. Data extraction will be done independently by two authors, and the consensus principle will be employed as a means of resolving existing disagreements among reviewers. If adequate data cannot be obtained from the full text, we will communicate with the corresponding author.

**Quality assessment of individual study and certainty assessment.** Our study will only include RCTs. Two authors will independently perform risk of bias assessment according to the second edition of the Cochrane Handbook for Systematic Reviews of Interventions of RCTs (RoB 2) [28–30]. The following six domains of bias will be appraised: random sequence generation, allocation concealment, blinding, missing data, selective reporting, and other biases. Trials with a high or unclear risk of bias in any of the first 3 components will be considered to have a high risk of bias. Any divergences in the risk of bias assessment will be resolved through consultation.

Two authors will respectively use the Grading of Recommendations Assessment, Development and Evaluation (GRADE) tool to evaluate the certainty of the evidence for our specific results [31]. The certainty of evidence will be categorized as either high, moderate, low, or very low. We will use GRADEpro Guideline Development Tool to conduct the GRADE assessments [32]. The process involves evaluating factors such as risk of bias, study design, inconsistency, indirectness, imprecision, effect size and publication bias to assess the confidence in the findings.

**Participants.** Clear inclusion and exclusion criteria for patient selection will be evaluated to ensure homogeneity of the study populations. In our study, we included only patients with

hip and proximal femur fractures and did not consider patients undergoing surgical treatment for bone tumors or femoral head necrosis. We will extract the patient's age, gender, ASA grade, injury site and comorbidity information for subsequent research analysis.

**Intervention.** All analgesia regimens for hip and proximal femur fractures will be considered, including fascia iliac compartment block (single-injection or continuous infusion) with or without systemic analgesia, and systemic analgesia alone. Details on the dose and frequency of analgesia, nerve block specifics (timing of FICB, technique: suprainguinal vs infrainguinal; method: landmark-based vs ultrasound-guided; provider: anesthesiologist vs emergency physician), type of local anesthetic (short-, medium-, or long-acting), surgical approach (minimally invasive vs open), anesthesia type (general vs intraspinal), and analgesia route (oral, intramuscular, or intravenous) will be extracted and analyzed.

**Outcome measures.** Regarding the primary outcome of pain score, we will extract pertinent details, including the time point of the pain assessment and the tools utilized for evaluation. Pain will be assessed at predefined time points: before treatment, immediately after treatment, and at 6, 24, and 48 hours post-treatment. If the pain scores of the included studies were inconsistent, we uniformly converted to a 10-point scale, where a lower pain score represented a lower degree of pain. The amount of opioid used in each study is uniformly converted to morphine.

Secondary outcomes will include success rate of nerve block, occurrence of adverse events (AEs), complications, patient satisfaction, and length of hospital stay, degree and duration of lower limb weakness. The success of nerve block includes clinical success and technical success. The former refers to decreased sensation in the skin area of the block area and decreased pain score. For technical success, this systematic review will extract data as defined in the included studies, such as successful deposition of the local anesthetic confirmed by ultrasound, loss of resistance, or other criteria specified by the authors [33]. AEs will be classified and categorized based on the definitions used in the included studies, and will be reported in terms of frequency and severity. AEs may include gastrointestinal, itchy skin, cardiovascular, or allergic reactions, among others. Complications such as infections, hematomas, or nerve damage will also be identified and categorized according to the criteria used in the original studies. Patient satisfaction will be evaluated at 24 and 48 hours post-treatment using the assessment tools specified in the included studies. Lower limb weakness will be assessed using the Lovett Muscle Strength Scale (0–5, with 5 indicating normal strength). Data from studies using alternative scales will be standardized for consistency.

The details of the outcome measures, including time points for pain evaluation, pain assessment tools, and criteria for identifying complications, will be systematically recorded and analyzed. Variations across studies will be addressed in the analysis, with subgroup analyses performed where applicable.

## Statistical analysis

**Pairwise meta-analysis.** First, when at least two studies evaluate the same measure, we will conduct a traditional paired meta-analysis using Rev Man 5.3 (Nordic Cochrane Centre, Denmark) to estimate the pooled odds ratio (OR) or standardized mean difference (SMD) and 95% confidence interval (95% CI). Heterogeneity among included studies will be tested using the $\chi^2$ method, and the $I^2$ statistic will be used to estimate the proportion of between-study variation to total variation [34]. If the $I^2$ statistic is > 50%, $p < 0.05$ indicates significant heterogeneity [34]. We will explain the sources of heterogeneity through pre-specified trial subgroups, graphical diagnostic tools, and sensitivity analyses, and then decide whether to use a fixed-effects model or a random-effects model for statistical analysis to produce more desirable results [34]. Multi-arms studies will be quantitatively included in pairwise meta-analysis based on specific comparisons.

**Network meta-analysis.** Following the traditional pairwise meta-analysis, a random effects network meta-analysis will be performed according to the methods described by Tian et al [35]. The initial values, automatically yield by WinBUGS 1.4.3 software (Imperial College School of Medicine, UK), will be applied to fit the model, and then we will conduct 70 000 iterations with a burn-in of 30 000 for each outcome and convergence. The evidence network diagrams will be drawn by Stata 13.0 (Stata Corp, Texas, USA) to show direct and indirect comparisons of the relationship between different analgesic methods. A cumulative ranking curve (SUCRA) will be plotted according to the primary indicators and adverse events to appraise benefit and risk of five different analgesic schemes [36]. Heterogeneity testing is the same as in traditional meta-analysis. For closed-loop networks, we will use an inconsistency model to evaluate the consistency between direct and indirect comparisons.

## Additional analyses

**Publication bias assessment.** Publication bias may not be detectable when the funnel plot represents a small number of studies. Therefore, we will use funnel plots to detect publication bias only when more than 10 studies are included [37]. We will also use the funnel plot to assess the small sample effect [38]. Furthermore, the objective diagnostic test will be conducted with Egger's test, and the Trim and fll method.

**Subgroup analysis.** To explore possible sources of heterogeneity or inconsistency, we will conduct subgroup analyses. These will include examining differences in the local anesthetic used for FICB, such as bupivacaine or ropivacaine, along with variations in dosage, concentration, and adjunctive medications like epinephrine or dexamethasone. We will also explore the timing of FICB administration (preoperative, intraoperative, or postoperative), the techniques and methods of FICB (suprainguinal vs infrainguinal approaches and landmark-based vs ultrasound-guided) and the influence of the operator, whether an anesthesiologist or emergency department physician. For systemic analgesia, we will investigate the impact of different regimens, comparing opioids to multimodal analgesia, and analyzing the dosing strategies. Patient characteristics, including age (e.g., < 65 vs. ≥ 65 years), fracture severity (stable vs. unstable), and potential ethnic or racial differences, will also be considered.

**Sensitivity analysis.** Finally, we will perform a series of sensitivity analyses to assess the robustness of our conclusions. These will include comparing the results using both fixed-effect and random-effect models to evaluate the impact of model choice on the findings. We will also exclude studies with a high risk of bias or those that do not meet predefined quality criteria to examine whether the inclusion of low-quality studies affects the results. Additionally, we will conduct a leave-one-out analysis, systematically excluding each study to determine whether any single study disproportionately influences the pooled effect size. These analyses will help ensure that our conclusions are not unduly influenced by study quality, methodological variations, or statistical assumptions.

## Discussion

Multiple systematic reviews have assessed analgesia for hip fracture patients, yet most focused on preoperative or prehospital analgesia, neglecting intraoperative and postoperative pain management [17,19,21]. There is no systematic review assessing the difference between single shot FICB and continuous FICB. Prior reviews mainly comprised low-quality literature, such as retrospective and non-randomized studies [17,39,40]. Previous systematic reviews did not include the entire age group, only adults [19], the elderly [41], or children [1]. Our comprehensive review analyzed the role of FICB across all age groups at all time points, being the first to assess the efficacy difference between single-shot and continuous FICB in hip fracture

analgesia. We only included high-quality RCTs to ensure robust evidence-based conclusions. To determine the effectiveness and safety of these analgesic methods, we will rank them based on major metrics and adverse events.

Despite the strengths of our review, there are several limitations that should be considered. First, there is heterogeneity in the analgesic techniques, including variations in the administration of fascia iliaca nerve block (FICB), such as differences in the type, dose, concentration, and adjuncts of anesthetics, as well as differences in FICB techniques and methods (suprainguinal vs. infrainguinal approach, landmark-based vs. ultrasound-guided). Additionally, there are variations in systemic analgesia regimens (opioids vs. multimodal approaches, dosage, and frequency). Second, the study population and design introduce potential confounding factors. Although we included studies covering all age groups, differences in baseline characteristics such as fracture severity and pre-fracture mobility may impact the results. Moreover, the variability in how outcomes are measured and reported (timing of pain assessments, definitions of adverse events) complicates direct comparisons. The limited follow-up duration in some studies also restricts our ability to assess long-term outcomes. Finally, while network meta-analysis allows for indirect comparisons of single-shot vs. continuous FICB, the validity of these comparisons depends on the assumption of transitivity across the network. Inconsistencies in study characteristics could weaken this assumption.

To address these limitations, our study will standardize the definitions of outcome measures and their time points. We will also conduct subgroup analyses to explore the heterogeneity across the studies. Additionally, we will carefully consider the design and baseline characteristics of each study to more accurately assess the effectiveness of different analgesic interventions.

The findings from our study have the potential to directly inform clinical practice and influence evidence-based decision-making. The identification of superior analgesic regimens can guide clinicians in selecting tailored strategies for pain management in patients with hip fractures. Specifically, our results could inform the revision of clinical guidelines by emphasizing the advantages of continuous FICB over single-shot FICB and systemic analgesia, depending on patient-specific factors.

In particular, FICB offers several advantages over systemic analgesia, especially in resource-limited settings where access to multimodal analgesia, advanced monitoring, and experienced pain management teams may be restricted. Compared to systemic opioids, FICB provides effective pain relief while reducing opioid consumption, thereby lowering the risk of opioid-related adverse effects such as respiratory depression, nausea, and delirium complications that can be particularly challenging to manage in under-resourced hospitals. Additionally, FICB can be performed with minimal equipment and training, making it a feasible option in rural or low-income settings where anesthesiologists may not always be available. The potential for prolonged pain relief with continuous FICB further supports its role in improving postoperative recovery and early mobilization, which are essential for reducing complications like deep vein thrombosis and pneumonia.

To ensure the findings translate into practice, we will engage with professional organizations and disseminate results through clinical forums and peer-reviewed journals. Incorporating these insights into guidelines can standardize care and optimize patient outcomes, particularly in resource-constrained settings where tailored strategies are crucial.

## Supporting information

**S1 Table. PRISMA-P 2015 checklist.**
(DOCX)

**S2 Table. The search strategy for PubMed.**
(DOCX)

## Acknowledgements

We thank Claire Lindow from University Hospitals Birmingham for polishing the language of our manuscript.

## Author contributions

**Conceptualization:** Jia-Xi Tang, Ling Wang, Shaojin Bu, Fang Gao, Tonny Veenith.

**Data curation:** Jia-Xi Tang, Ling Wang, Shaojin Bu, Wallisa Roberts, Fang Gao, Rajneesh Sachdeva.

**Formal analysis:** Raja V Lakshmanan, Rajneesh Sachdeva.

**Funding acquisition:** Jia-Xi Tang, Shaojin Bu.

**Investigation:** Shaojin Bu, Wallisa Roberts, Narcis Ungureanu, Ansar Mahmood.

**Methodology:** Jia-Xi Tang, Ling Wang, Shaojin Bu, Wallisa Roberts, Tonny Veenith, Rajneesh Sachdeva.

**Project administration:** Fang Gao.

**Software:** Jia-Xi Tang, Ling Wang, Tonny Veenith.

**Supervision:** Fang Gao.

**Validation:** Jia-Xi Tang, Ling Wang, Shaojin Bu, Wallisa Roberts, Narcis Ungureanu, Ansar Mahmood, Fang Gao, Raja V Lakshmanan, Tonny Veenith, Rajneesh Sachdeva.

**Writing – original draft:** Jia-Xi Tang, Ling Wang.

**Writing – review & editing:** Ling Wang, Narcis Ungureanu, Tonny Veenith, Rajneesh Sachdeva.

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
