## [Decision Letter · Decision Letter 0]

31 May 2024

Dear Dr. Tang,

The required changes mentioned by reviewers 1 and 2 are detailed below. The introduction, method, and discussion sections need special attention and major revisions. We can then assess the paper for acceptance if it strictly adheres to the journal's policies and guidelines.

We look forward to receiving your revised manuscript.

Kind regards,

Muhammad Waqas Khan

Academic Editor

PLOS ONE

Additional Editor Comments:

Reviewers' comments:

Reviewer's Responses to Questions

**Comments to the Author**

1. Does the manuscript provide a valid rationale for the proposed study, with clearly identified and justified research questions?

Reviewer #1: Partly

Reviewer #2: Partly

2. Is the protocol technically sound and planned in a manner that will lead to a meaningful outcome and allow testing the stated hypotheses?

Reviewer #1: Yes

Reviewer #2: Yes

3. Is the methodology feasible and described in sufficient detail to allow the work to be replicable?

Reviewer #1: Yes

Reviewer #2: No

4. Have the authors described where all data underlying the findings will be made available when the study is complete?

Reviewer #1: No

Reviewer #2: Yes

5. Is the manuscript presented in an intelligible fashion and written in standard English?

Reviewer #1: Yes

Reviewer #2: Yes

You may also provide optional suggestions and comments to authors that they might find helpful in planning their study.

Reviewer #1: This systematic review protocol and Network meta-analysis evaluates traditional systemic analgesia interventions with continuous or single shot anesthetic block of the fascia iliaca in patients with fractures of the hip or proximal femur regarding pain and the use of opioids. The methodology follows PRISMA recommendations.

I suggest that some points be reviewed by the authors:

- Introduction – The authors describe that although several studies, including systematic reviews, demonstrate that Fascia Iliaca compartment block is effective and reduces the use of opioids in patients with fractures of the proximal femur, this is correct. “However, Dai et.al[20] and Smith et.al[21] indicated that FICB is not superior to placebo for patients undergoing hip surgeries”, these two studies refer to arthroscopy and primary hip arthroplasty, surgeries that do not are the subject of this review and therefore should not be used as a rationale for preparing this protocol.

- Study Selection –

linhas 112-114-

Language: “Considering that there are no translators proficient in other languages, only full-text articles published in English or Chinese”. I suggest that there is no language restriction, for example, in the study by Slade (2023) two studies published in French were included.

Line 119- “ Exclusion criteria: (i) replicated and low-quality studies” – I suggest describing the criteria adopted to define which low-quality studies will be excluded.

- Plos data Policy - According to the Plos data policy, authors are required to describe where all data underlying the findings of their manuscript will be made available (database or public repository), without restrictions.

Reviewer #2: The respected authors have presented a Pilot study to assess the pain management strategies following a hip fracture. The authors tend to investigate the impact of the single and continuous fascia Iliaca Compartment Block technique in comparison to the conventional systemic analgesics, employing a Bayesian Network meta-analysis method. The manuscript will address a major clinical problem.

However, the author should address the following issues to improve the quality of the manuscript.

Introduction:

Lines 47-48: The author states the possible prevalence projection of hip fracture; however, the author needs to add statistical evidence from the listed references to provide support for their statement.

Lines 48-50: the author cited the 2011 Association of Anaesthetists, the author should cite the recent guideline published in 2020.

Line 68-69: the author should be emphasizing the hypothesis and not the methodology in the introduction. The statement for the Bayesian NMA and indirect comparison is more suitable in the method section and is irrelevant to the introduction.

Line 52-54: the statement is unclear please rewrite the statement for more clarity.

Line 72-73: the author presents their hypothesis in the former sentence, however, in lines 72-73, the author should state that they will assess the duration of lower limb weakness as a secondary outcome, however, the written statement looks like the author is quite sure about the possible outcome and its risk regarding the weakness. This statement is not relevant in writing an introduction.

Methods:

Line 91-92: the author included only the Fascia iliaca block as an intervention in the search strategy. Similarly, table S1 also shows the strategy only restricted to the following keywords. In contrast, the network diagram shows systemic analgesics, similarly, the inclusion criteria also include systemic analgesics in lines 107-108. Therefore, systemic analgesia should also be included in the search strategy to provide the rank-based analysis of all the interventions considering the author is planning to conduct a network meta-analysis. However, according to the hypothesis, the author seems only interested in the single and continuous FICB so a meta-analysis or a network analysis only with the three interventions: single, continuous FICB and placebo should be conducted, along with an update of the illustrated network plot will be required. The author needs to refine and redefine their methodology, search strategy, PICO and hypothesis regarding the inclusion or exclusion of systemic analgesics.

Line 164: the statement is unclear.

Discussion

Line 181: the author claims the novelty of the study however, a previously published network meta-analysis entitled, “Hayashi M, Yamamoto N, Kuroda N, Kano K, Miura T, Kamimura Y, Shiroshita A. Peripheral Nerve Blocks in the Preoperative Management of Hip Fractures: A Systematic Review and Network Meta-Analysis. Ann Emerg Med. 2024 Jun;83(6):522-538. doi: 10.1016/j.annemergmed.2024.01.024. Epub 2024 Feb 22. PMID: 38385910” highlighted a similar subject. The author should be describing more reasons regarding the uniqueness of the study.

**Do you want your identity to be public for this peer review?** For information about this choice, including consent withdrawal, please see our Privacy Policy

Reviewer #1: **Yes: ** JOAO CARLOS BELLOTI

Reviewer #2: **Yes: ** Shayan Ali Irfan

---

## [Author Response · Author response to Decision Letter 1]

1 Jul 2024

Jiaxi Tang MD. PhD

Department of Anesthesiology, Chongqing Key Laboratory of Translational Research for Cancer Metastasis and Individualized Treatment, Chongqing University Cancer Hospital,

Chongqing 400030, China

E-mail: tangjiaxi1029@126.com

June 26, 2024

RE: Manuscript ID: PONE-D-23-33546

Dear Editor，

We would like to thank the editor for giving us a chance to resubmit the paper, and also thank the reviewer for giving us constructive suggestions which would help us both in English and in depth to improve the quality of the paper. Here we submit a new version of our manuscript with the title “Comparison of traditional systemic analgesic, Single shot or continuous Fascia Iliaca Compartment Block for pain management in patients with hip or Proximal Femoral Fractures: A protocol for systematic review and network meta-analysis”, which has been modified according to the reviewers’ suggestions. Efforts were also made to correct the mistakes and improve the English of the manuscript. We highlight all the changes using a track change function in the revised manuscript.

Sincerely yours,

Jiaxi Tang MD. PhD

The following is a point-by-point response to the editor and reviewers’ comments.

Editor:

General comments:

As there are a number of omissions in the design of this study and written English is not academic standard, the value of this topic under review is questionable.

Answer: Thank you for comments and guidance on the paper, and thanks for giving us the opportunity to revise it. We have revised our manuscript based on comments from editors and reviewers to make it more readable and clear.

Specific Comments:

1. Abstract/ Introduction-content

No read

Answer: Thank you for your comments. We will make revisions based on comments from editor and reviewers. Please review it again after the revision is completed. Thank you. (Line 32-36)

2. Methods: General

-It would be better to have two subtitles as “Identification and selection of studies” and “Assessment of characteristics of studies”.

-Information is messy, so it makes readers feel difficult to read.

Answer: Thank you for your enlightening advice, which can improve the quality and readability of our manuscripts. We have readjusted the order of the methods section, adding two subtitles: "Identification and selection of studies" and "Assessment of characteristics of studies". (Line 90; Line 133).

3. Methods: Identification and selection of studies

Information is messy. Inclusion criteria should be following as Design (missing what kind of design would be included), Participants (not clear), Outcome measures (no enough information)

Searches

For each database, it should be as example like: Embase (1947 to June 2023). This is missing

Inclusion criteria: this should be part of Identification and selection of studies

Authors should tidy up information, perhaps using a table to cover all area, such as Design, Participants, Intervention, Outcome measures, Comparisons.

Answer: Thank you for your comments and suggestions. We have updated the order of inclusion criteria in our manuscript. In addition, we have added information for parts of the manuscript that are not explicitly described, and have increased the specific search intervals for each databases. (Line 93-97; line 114-123).

4. Methods: Assessment of characteristics of studies

There should be more points in this part. It would be better to have this headings and follow some subheading as below:

Quality: missing

Participants: missing

Intervention: messing

Outcome measures: missing

Answer: Thank you for your enlightening comments, which will make our manuscript more organized and legible. We have revised this part according to your comments. (Line 153-180).

5. Results/ Discussion/ Conclusion: context.

As there are so many questions in the method, results, discussion and conclusion are questionable.

Answer: Thank you for your reminder and commets. We have revised the discussion section accordingly. (Line 219-229).

6. Readability and style.

This paper is not easy to read, because the structure of written information.

Answer: Thank you for your comments. We have revised the structure of the manuscript in response to comments from editors and reviewers. (Line 90-217).

Reviewer #1:

General comments:

This systematic review protocol and Network meta-analysis evaluates traditional systemic analgesia interventions with continuous or single shot anesthetic block of the fascia iliaca in patients with fractures of the hip or proximal femur regarding pain and the use of opioids. The methodology follows PRISMA recommendations.

I suggest that some points be reviewed by the authors:

- Introduction – The authors describe that although several studies, including systematic reviews, demonstrate that Fascia Iliaca compartment block is effective and reduces the use of opioids in patients with fractures of the proximal femur, this is correct. “However, Dai et.al[20] and Smith et.al[21] indicated that FICB is not superior to placebo for patients undergoing hip surgeries”, these two studies refer to arthroscopy and primary hip arthroplasty, surgeries that do not are the subject of this review and therefore should not be used as a rationale for preparing this protocol.

- Study Selection –

linhas 112-114-

Language: “Considering that there are no translators proficient in other languages, only full-text articles published in English or Chinese”. I suggest that there is no language restriction, for example, in the study by Slade (2023) two studies published in French were included.

Line 119- “ Exclusion criteria: (i) replicated and low-quality studies” – I suggest describing the criteria adopted to define which low-quality studies will be excluded.

- Plos data Policy - According to the Plos data policy, authors are required to describe where all data underlying the findings of their manuscript will be made available (database or public repository), without restrictions.

Answer: Thank you for your comments and reminder. We have revised our manuscript accordingly. I would also like to thank you for your valuable comments on the paper, which have made a great contribution to improving the quality and accuracy of the paper.

Specific Comments:

1. Introduction – The authors describe that although several studies, including systematic reviews, demonstrate that Fascia Iliaca compartment block is effective and reduces the use of opioids in patients with fractures of the proximal femur, this is correct. “However, Dai et.al[20] and Smith et.al[21] indicated that FICB is not superior to placebo for patients undergoing hip surgeries”, these two studies refer to arthroscopy and primary hip arthroplasty, surgeries that do not are the subject of this review and therefore should not be used as a rationale for preparing this protocol.

Answer: Thank you for your comments on the paper, which are of great help in improving the rigor of our manuscript. We have substituted a suitable reference to elucidate the rationale underlying our research. (Line 63-69).

2. Study Selection –

linhas 112-114-

Language: “Considering that there are no translators proficient in other languages, only full-text articles published in English or Chinese”. I suggest that there is no language restriction, for example, in the study by Slade (2023) two studies published in French were included.

Answer: Thank you for the comments on the paper. We have removed the restrictions on the language of publication. Indeed, limiting the language of the included study is not conducive to a comprehensive analysis of our topic. Thank you again for your valuable advice. (Line 123).

3. Line 119- “ Exclusion criteria: (i) replicated and low-quality studies” – I suggest describing the criteria adopted to define which low-quality studies will be excluded.

Answer: Thank you for your constructive comments. This is essential to clarify the inclusion and exclusion criteria for our studies and to facilitate replication of our studies by other researchers. We have established the criteria for low-quality studies based on previous literature. (Line 129).

4. Plos data Policy - According to the Plos data policy, authors are required to describe where all data underlying the findings of their manuscript will be made available (database or public repository), without restrictions.

Answer: Thank you for your comments and kind reminder. We will provide a complete Data Availability Statement in the submission form. Thanks again.

Reviewer #2:

General comments:

The respected authors have presented a Pilot study to assess the pain management strategies following a hip fracture. The authors tend to investigate the impact of the single and continuous fascia Iliaca Compartment Block technique in comparison to the conventional systemic analgesics, employing a Bayesian Network meta-analysis method. The manuscript will address a major clinical problem.

However, the author should address the following issues to improve the quality of the manuscript.

Introduction:

Lines 47-48: The author states the possible prevalence projection of hip fracture; however, the author needs to add statistical evidence from the listed references to provide support for their statement.

Lines 48-50: the author cited the 2011 Association of Anaesthetists, the author should cite the recent guideline published in 2020.

Line 68-69: the author should be emphasizing the hypothesis and not the methodology in the introduction. The statement for the Bayesian NMA and indirect comparison is more suitable in the method section and is irrelevant to the introduction.

Line 52-54: the statement is unclear please rewrite the statement for more clarity.

Line 72-73: the author presents their hypothesis in the former sentence, however, in lines 72-73, the author should state that they will assess the duration of lower limb weakness as a secondary outcome, however, the written statement looks like the author is quite sure about the possible outcome and its risk regarding the weakness. This statement is not relevant in writing an introduction.

Methods:

Line 91-92: the author included only the Fascia iliaca block as an intervention in the search strategy. Similarly, table S1 also shows the strategy only restricted to the following keywords. In contrast, the network diagram shows systemic analgesics, similarly, the inclusion criteria also include systemic analgesics in lines 107-108. Therefore, systemic analgesia should also be included in the search strategy to provide the rank-based analysis of all the interventions considering the author is planning to conduct a network meta-analysis. However, according to the hypothesis, the author seems only interested in the single and continuous FICB so a meta-analysis or a network analysis only with the three interventions: single, continuous FICB and placebo should be conducted, along with an update of the illustrated network plot will be required. The author needs to refine and redefine their methodology, search strategy, PICO and hypothesis regarding the inclusion or exclusion of systemic analgesics.

Line 164: the statement is unclear.

Discussion

Line 181: the author claims the novelty of the study however, a previously published network meta-analysis entitled, “Hayashi M, Yamamoto N, Kuroda N, Kano K, Miura T, Kamimura Y, Shiroshita A. Peripheral Nerve Blocks in the Preoperative Management of Hip Fractures: A Systematic Review and Network Meta-Analysis. Ann Emerg Med. 2024 Jun;83(6):522-538. doi: 10.1016/j.annemergmed.2024.01.024. Epub 2024 Feb 22. PMID: 38385910” highlighted a similar subject. The author should be describing more reasons regarding the uniqueness of the study.

Answer: Thanks for your comments and valuable advice. Thank you for your comments on the paper, which are very helpful in improving the quality of our manuscript. We have revised our paper according to your insightful advice.

Specific Comments:

1. Introduction: Lines 47-48: The author states the possible prevalence projection of hip fracture; however, the author needs to add statistical evidence from the listed references to provide support for their statement.

Answer: Thank you for the comments on the paper. We have revised the manuscript based on the reviewers’ suggestions. (Line 48-51).

2. Lines 48-50: the author cited the 2011 Association of Anaesthetists, the author should cite the recent guideline published in 2020.

Answer: Thank you for your kind comments. We have updated the reference. (Line 54, 259-261).

3. Line 68-69: the author should be emphasizing the hypothesis and not the methodology in the introduction. The statement for the Bayesian NMA and indirect comparison is more suitable in the method section and is irrelevant to the introduction.

Answer: Thank you for your pertinent comments, which are of great help in improving the quality of our manuscript. We have removed inappropriate statements from the introduction section. (Line 74-77).

4. Line 52-54: the statement is unclear please rewrite the statement for more clarity.

Answer: Thank you for your comments, which are very important to improve the readability of the article. We have revised the manuscript according to the comments of reviewer. (Line 55-58).

5. Line 72-73: the author presents their hypothesis in the former sentence, however, in lines 72-73, the author should state that they will assess the duration of lower limb weakness as a secondary outcome, however, the written statement looks like the author is quite sure about the possible outcome and its risk regarding the weakness. This statement is not relevant in writing an introduction.

Answer: Thank you for your comments and advice. We apologize for any confusion caused by the unclear expression in our manuscript. In fact, our hypothesis encompasses both the analgesic efficacy and safety of FICB, and we have integrated these two aspects. Furthermore, in our study, we will evaluate the duration of lower limb weakness as a secondary outcome. (Line 74-79, 123, 178, 179).

6. Methods:

Line 91-92: the author included only the Fascia iliaca block as an intervention in the search strategy. Similarly, table S1 also shows the strategy only restricted to the following keywords. In contrast, the network diagram shows systemic analgesics, similarly, the inclusion criteria also include systemic analgesics in lines 107-108. Therefore, systemic analgesia should also be included in the search strategy to provide the rank-based analysis of all the interventions considering the author is planning to conduct a network meta-analysis. However, according to the hypothesis, the author seems only interested in the single and continuous FICB so a meta-analysis or a network analysis only with the three interventions: single, continuous FICB and placebo should be conducted, along with an update of the illustrated network plot will be required. The author needs to refine and redefine their methodology, search strategy, PICO and hypothesis regarding the inclusion or exclusion of systemic analgesics.

Answer: Thank you for your valuable suggestions and reminders, which are of great help to improve the logic and readability of the article. We carefully checked the inclusion and exclusion criteria of our manuscript, revised the contradictions, and updated table S2 and Fig 2. (Line101, 102, 117-119, also see table S2 and Fig 2).

7. Line 164: the statement is unclear.

Answer: Thank you for your comments. We have removed the ambiguous expression. (Line 209).

8. Discussion

Line 181: the author claims the novelty of the study however, a previously published network meta-analysis entitled, “Hayashi M, Yamamoto N, Kuroda N, Kano K, Miura T, Kamimura Y, Shiroshita A. Peripheral Nerve Blocks in the Preoperative Management of Hip Fractures: A Systematic Review and Network Meta-Analysis. Ann Emerg Med. 2024 Jun;83(6):522-538. doi: 10.1016/j.annemergmed.2024.01.024. Epub 2024 Feb 22. PMID: ” highlighted a similar subject. The author should be describing more reasons regarding the uniqueness of the study.

Answer: T

---

## [Decision Letter · Decision Letter 1]

15 Nov 2024

Dear Dr. Tang,

Thank you for submitting your manuscript to PLOS ONE. After careful consideration, we feel that it has merit but does not fully meet PLOS ONE’s publication criteria as it currently stands. Therefore, we invite you to submit a revised version of the manuscript that addresses the points raised during the review process.

In Introduction:

Provide a brief background on the systemic analgesics and fascia iliaca nerve blocks as how they provide analgesia for hip or proximal femoral fractures.Comment on what type of pain and analgesia duration the investigators are attempting to address.It’s a well-known phenomenon that pain management for joint surgery requires multimodal analgesia. So, what is the rationale behind comparing systemic analgesia vs peripheral nerve blocks?What is the rationale behind investigating a single shot vs a continuous nerve block?

In materials and methods:

Comment on how studies including shaft and distal femoral fractures will be reliably excluded during the search.Inter variability between the types of fascia iliaca blocks and the providers who perform them have a significant influence on outcomes, hence will generically including all under one umbrella be a meaningful investigation?Define “abstracts with sufficient data”.Shouldn’t the fascia iiaca block studies either be inclusive or exclusive of systemic analgesia as they will eventually impact the final outcomes.Clarify the duration when the primary outcomes are going to be investigated.Clarify the outcome measure technical success. How will the successful injection of local anesthetic be determined?Define the secondary outcomes such as adverse events, complications and patient satisfaction

publication criteria  and not, for example, on novelty or perceived impact.

We look forward to receiving your revised manuscript.

Kind regards,

Vendhan Ramanujam, M.B.B.S, M.D.

Academic Editor

PLOS ONE

Reviewers' comments:

Reviewer's Responses to Questions

**Comments to the Author**

1. Does the manuscript provide a valid rationale for the proposed study, with clearly identified and justified research questions?

Reviewer #3: Yes

2. Is the protocol technically sound and planned in a manner that will lead to a meaningful outcome and allow testing the stated hypotheses?

Reviewer #3: Yes

3. Is the methodology feasible and described in sufficient detail to allow the work to be replicable?

Reviewer #3: Yes

4. Have the authors described where all data underlying the findings will be made available when the study is complete?

Reviewer #3: Yes

5. Is the manuscript presented in an intelligible fashion and written in standard English?

Reviewer #3: Yes

You may also provide optional suggestions and comments to authors that they might find helpful in planning their study.

Reviewer #3: 1. Studies

Line 117-118 mentions the inclusion of all analgesia regimens, including fascia iliaca compartment block (single-injection or continuous infusion) and systemic analgesia alone. Could you clarify whether your review will include only studies comparing single-shot vs. continuous FICB, or if studies comparing FICB to placebo or systemic analgesia to placebo will also be included? If so, how will this align with your hypothesis that continuous FICB may prolong analgesic effects and reduce opioid use compared to single-shot or systemic analgesia?

2. Data Collection

Could you discuss potential challenges in collecting data from the included studies, especially considering variations in how different analgesic techniques are administered (e.g., differences in dosage, technique, or patient populations)? This would help clarify the limitations of the meta-analysis.

3. Outcome Measures

While you mention outcome measures, could you provide more details on how pain scores, mobility, and adverse events will be standardized across studies to manage potential heterogeneity?

4. Discussion

How do you envision translating the findings from this study into clinical practice, particularly with respect to updating pain management guidelines for hip fractures?

**Do you want your identity to be public for this peer review?** For information about this choice, including consent withdrawal, please see our Privacy Policy

Reviewer #3: **Yes: ** Sujatha Baddam

---

## [Author Response · Author response to Decision Letter 2]

8 Dec 2024

RE: Manuscript ID: PONE-D-23-33546R1

Dear Editor，

We would like to thank the editor for giving us a chance to resubmit the paper, and also thank the reviewer for giving us constructive suggestions which would help us in depth to improve the quality of the paper. Here we submit a new version of our manuscript with the title “Comparison of traditional systemic analgesic, Single shot or continuous Fascia Iliaca Compartment Block for pain management in patients with hip or Proximal Femoral Fractures: A protocol for systematic review and network meta-analysis”, which has been modified according to the reviewer and editor’s suggestions. We highlight all the changes using a track change function in the revised manuscript.

Sincerely yours,

Jiaxi Tang MD. PhD

The following is a point-by-point response to the editor and reviewer’s comments.

Editor:

General comments:

Although your corrections to the previous reviews have improved the manuscript, it still needs attention to several details and clarifications.

Answer: Thank you for comments and guidance on the paper, and thanks for giving us the opportunity to revise it. We have revised our manuscript based on comments from editor and reviewer to make it more readable and clear.

Specific Comments:

1. In Introduction:

1) Provide a brief background on the systemic analgesics and fascia iliaca nerve blocks as how they provide analgesia for hip or proximal femoral fractures.

Answer: Thank you for your valuable advice. Our previous manuscript did not provide a brief background on the systemic analgesic regimen and fascia iliaca nerve blocks for hip or proximal femoral fractures analgesia, which may be unfriendly to non-specialists and add to the difficulty and confusion of reading. We have added this section, thanks again. (Line 60-69)

2) Comment on what type of pain and analgesia duration the investigators are attempting to address.

Answer: Thank you for your insightful comment regarding the type of pain and analgesia duration addressed in our study. We included patients with hip fractures regardless of whether they underwent surgery to reflect the diversity of real-world clinical scenarios. Our study did not specifically focus on preoperative, postoperative, or admission-related pain as our primary goal was to evaluate the overall efficacy of analgesic modalities in managing acute pain during hospitalization. In addition, we have included a specific time frame for pain assessment in the methodological section to enhance the clarity of our research.(Line 85-88, 184-186)

To address this complexity, our study protocol includes a predefined subgroup analysis based on the timing of nerve block administration. This analysis will allow us to explore the impact of nerve block timing on analgesic outcomes and provide more nuanced insights into pain management strategies across different stages of care.(Line 240-252)

3) It’s a well-known phenomenon that pain management for joint surgery requires multimodal analgesia. So, what is the rationale behind comparing systemic analgesia vs peripheral nerve blocks?

Answer: Thank you for highlighting the importance of multimodal analgesia in joint surgery pain management. We agree that such approaches are essential. The rationale for our comparison between systemic analgesia and peripheral nerve blocks is to explore their individual roles within multimodal strategies. Systemic analgesics, though effective and widely accessible, can lead to dose-dependent side effects. Peripheral nerve blocks, on the other hand, offer localized pain relief with fewer systemic effects but require specialized resources.

By comparing these methods, our study aims to provide evidence that helps clinicians optimize multimodal analgesia, balancing efficacy, safety, and resource considerations to meet the needs of diverse patient populations.(Line 85-88)

4) What is the rationale behind investigating a single shot vs a continuous nerve block?

Answer: Thank you for raising this important point. Single-shot nerve blocks provide effective pain relief; however, their duration is inherently limited, often necessitating supplemental analgesia, such as systemic opioids, to manage breakthrough pain. Additionally, continuous FICB requires specialized equipment and techniques and carries risks such as weakened quadriceps strength, delayed recovery, and prolonged hospital stays. These limitations are the key reasons for investigating the comparative efficacy of single-shot versus continuous nerve blocks. Our study aims to evaluate how these approaches address acute pain in hip fracture patients and inform optimal analgesic strategies that balance efficacy, safety, and resource utilization. (Line 81-83)

2. In materials and methods:

1) Comment on how studies including shaft and distal femoral fractures will be reliably excluded during the search.

Answer: Thank you for your question regarding the exclusion of shaft and distal femoral fractures during the search. To ensure reliable exclusion, we designed our search strategy to explicitly differentiate between proximal femoral fractures and other types of femoral fractures. Specifically, as shown in Supplementary Table S2, we used the MeSH term "Femoral Fractures"[Mesh] and excluded "Femoral Fractures, Distal"[Mesh] with the NOT operator.(S2 Table) We also made changes in the manuscript. (Line 110-111)

2) Inter variability between the types of fascia iliaca blocks and the providers who perform them have a significant influence on outcomes, hence will generically including all under one umbrella be a meaningful investigation?

Answer: Thank you for highlighting the potential inter-variability between different types of fascia iliaca blocks (FIB) and the providers who perform them. We fully agree that these factors can significantly influence analgesic outcomes and warrant careful consideration.

To address this, our study protocol includes predefined subgroup analyses that will stratify outcomes based on the specific FIB techniques (e.g., suprainguinal vs infrainguinal approaches), methods (e.g., landmark-based vs ultrasound-guided), and the provider performing the block (e.g., anesthesiologists vs emergency department physicians). By incorporating these subgroup analyses, we aim to better understand the impact of these variables on the efficacy of FIB and to provide more nuanced insights for clinical practice.

We believe this approach balances the need for capturing real-world diversity with the importance of accounting for key sources of variability, ensuring the robustness and applicability of our findings. (Line 176-181, 240-252)

3) Define “abstracts with sufficient data”.

Answer: Thank you for pointing out the need to define "abstracts with sufficient data." In our study, this term refers to abstracts that provide adequate information to assess eligibility and extract key outcomes, such as patient population, intervention (fascia iliaca nerve block), comparator (systemic analgesia), and primary or secondary outcomes related to analgesic efficacy or safety.

This definition ensures that only abstracts with the necessary details to contribute to our analysis are included, while abstracts lacking critical information (e.g., no description of the intervention or outcomes) are excluded. We have now included this definition in the manuscript for clarity. (Line 125-129)

4) Shouldn’t the fascia iiaca block studies either be inclusive or exclusive of systemic analgesia as they will eventually impact the final outcomes.

Answer: Thank you for your thoughtful comment regarding the inclusion or exclusion of systemic analgesia in fascia iliaca block (FICB) studies. We agree that systemic analgesia can have a significant impact on the final outcomes, and it is important to consider how its use may influence the effectiveness of FICB.

In our study, we have decided to include studies regardless of whether systemic analgesia was used in combination with FICB. This approach reflects real-world clinical practice, where FICB is often used alongside systemic analgesia to provide more comprehensive pain management.

To compare multiple analgesic treatments in this study, we plan to conduct a network meta-analysis and plot cumulative ranking curves (SUCRA) based on key outcomes and adverse events to assess the benefits and risks of various analgesic regimens. (Line 228-230)

5) Clarify the duration when the primary outcomes are going to be investigated.

Answer: Thank you for your comment regarding the duration for assessing the primary outcomes. The primary outcome in our study is pain score, which will be measured at predefined time points based on clinical relevance. Specifically, Pain will be assessed at predefined time points: before treatment, immediately after treatment, and at 6, 24, and 48 hours post-treatment, which are chosen to capture both the immediate and long-term effects of the analgesic treatments.

We believe that these time points will offer a comprehensive understanding of the analgesic efficacy and pain relief provided by different interventions for acute pain in hospitalized patients. (Line 183-188)

6) Clarify the outcome measure technical success. How will the successful injection of local anesthetic be determined?

Answer: Thank you for your comment. In this systematic review, we define technical success based on criteria commonly reported in the included studies. Specifically, technical success is confirmed by one or more of the following:

Ultrasound Confirmation: Successful visualization of the local anesthetic within the fascia iliaca compartment.

Loss of resistance: When the tip of the needle passes through the fascia lata and the fascia iliaca, a loss of resistance is felt, as commonly described in the literature.

Or other criteria specified by the authors (Line 193-195)

7) Define the secondary outcomes such as adverse events, complications and patient satisfaction

Answer: Thank you for your insighted suggestion. Defining secondary indicators in advance will make the evaluation of secondary indicators in this system evaluation more transparent and targeted. We have revised it according to your comments. (Line 189-203)

Reviewer #3:

General comments:

1. Does the manuscript provide a valid rationale for the proposed study, with clearly identified and justified research questions?

Reviewer #3: Yes

2. Is the protocol technically sound and planned in a manner that will lead to a meaningful outcome and allow testing the stated hypotheses?

Reviewer #3: Yes

3. Is the methodology feasible and described in sufficient detail to allow the work to be replicable?

Reviewer #3: Yes

4. Have the authors described where all data underlying the findings will be made available when the study is complete?

Reviewer #3: Yes

5. Is the manuscript presented in an intelligible fashion and written in standard English?

Reviewer #3: Yes

Answer: Thank you for your comments and recognition of our research.

Specific Comments:

1. Studies

Line 117-118 mentions the inclusion of all analgesia regimens, including fascia iliaca compartment block (single-injection or continuous infusion) and systemic analgesia alone. Could you clarify whether your review will include only studies comparing single-shot vs. continuous FICB, or if studies comparing FICB to placebo or systemic analgesia to placebo will also be included? If so, how will this align with your hypothesis that continuous FICB may prolong analgesic effects and reduce opioid use compared to single-shot or systemic analgesia?

Answer: Thank you for your insightful comments and for the opportunity to clarify our study design. Regarding your query about the inclusion criteria for our review (Lines 117-118), we confirm that our study will include comparisons among the following analgesic regimens, as illustrated in Figure 2: 1. Single-shot FICB (± systemic analgesia) 2. Continuous FICB (± systemic analgesia) 3. Systemic analgesia alone (e.g., opioids or multimodal analgesia)

However, we will not include studies comparing systemic analgesia to placebo studies. In cases of hip fracture, the absence of analgesia is neither ethically permissible nor practically feasible; thus, such studies are unlikely to exist or be eligible for inclusion.

Furthermore, we acknowledge that studies directly comparing single-shot FICB to continuous FICB are limited. To address this, we plan to use a network meta-analysis (NMA) approach. This method enables indirect comparisons through a connected network of studies, incorporating data from multiple analgesic regimens. By leveraging this framework, we aim to comprehensively assess the relative efficacy and safety of single-shot versus continuous FICB, even in the absence of abundant direct comparative studies.

We appreciate your thoughtful feedback, which has helped us to clarify and strengthen the description of our study methodology. Please let us know if further clarification is needed. (Line 221-232, Figure2)

2. Data Collection

Could you discuss potential challenges in collecting data from the included studies, especially considering variations in how different analgesic techniques are administered (e.g., differences in dosage, technique, or patient populations)? This would help clarify the limitations of the meta-analysis.

Answer: Thank you for raising this critical point regarding potential challenges in data collection and the implications for our meta-analysis. Variations in how analgesic techniques are administered, including systemic analgesia regimens, indeed present challenges for synthesizing data.

We have revised the discussion section and the method section to address the limitations of the study and solutions to the problem.(Line 240-252, 265-285).

3. Outcome Measures

While you mention outcome measures, could you provide more details on how pain scores, mobility, and adverse events will be standardized across studies to manage potential heterogeneity?

Answer: Thank you for your constructive comments. This is essential to clarify the outcome measures criteria for our studies an

---

## [Decision Letter · Decision Letter 2]

4 Feb 2025

Dear Dr. Tang,

As the Jadad score is only used for inclusion decisions and not for quality appraisal, it should be deleted from the abstractPlease make sure that in the abstract and in the text, you refer to the appraisal tool as ROB 2 to avoid misunderstandingsPlease mention the use of the GRADE approach in the abstract. In the text, please make sure that you use an appropriate original reference for GRADE and describe the instrument and the process in some more detail. For example, are you planning to use GRADEpro? Please mind that results of the GRADE assessments are “certainty (not grade) of the evidence”For screening of results, EndNote seems not sufficient, please state the use of an appropriate software as e.g. Covidence, Rayyan or EppiReviewer

We look forward to receiving your revised manuscript.

Kind regards,

Sascha Köpke

Academic Editor

PLOS ONE

Journal Requirements:

Reviewers' comments:

Reviewer's Responses to Questions

**Comments to the Author**

1. Does the manuscript provide a valid rationale for the proposed study, with clearly identified and justified research questions?

Reviewer #3: Yes

2. Is the protocol technically sound and planned in a manner that will lead to a meaningful outcome and allow testing the stated hypotheses?

Reviewer #3: Yes

3. Is the methodology feasible and described in sufficient detail to allow the work to be replicable?

Reviewer #3: Yes

4. Have the authors described where all data underlying the findings will be made available when the study is complete?

Reviewer #3: Yes

5. Is the manuscript presented in an intelligible fashion and written in standard English?

Reviewer #3: Yes

You may also provide optional suggestions and comments to authors that they might find helpful in planning their study.

Reviewer #3: Subgroup Analysis

Consider including a sensitivity analysis to assess the robustness of findings.

Discussion

More emphasis should be placed on the practical implications of using FICB over systemic analgesia, particularly in resource-limited settings.

**Do you want your identity to be public for this peer review?** For information about this choice, including consent withdrawal, please see our Privacy Policy

Reviewer #3: **Yes: ** Sujatha Baddam

---

## [Author Response · Author response to Decision Letter 3]

6 Feb 2025

RE: Manuscript ID: PONE-D-23-33546R2

Dear Editor，

We would like to thank the editor for giving us a chance to resubmit the paper, and also thank the reviewer for giving us constructive suggestions which would help us in depth to improve the quality of the paper. Here we submit a new version of our manuscript with the title “Comparison of traditional systemic analgesic, Single shot or continuous Fascia Iliaca Compartment Block for pain management in patients with hip or Proximal Femoral Fractures: A protocol for systematic review and network meta-analysis”, which has been modified according to the reviewer and editor’s suggestions. We highlight all the changes using a track change function in the revised manuscript.

Sincerely yours,

Jiaxi Tang MD. PhD

The following is a point-by-point response to the editor and reviewer’s comments.

Editor:

Specific comments:

1. As the Jadad score is only used for inclusion decisions and not for quality appraisal, it should be deleted from the abstract

Answer: Thank you for your valuable feedback. We understand that the Jadad score is primarily used for the inclusion decision rather than as a comprehensive quality appraisal tool. To avoid any confusion, we will revise the manuscript according to your suggestion. (Line 39)

2. Please make sure that in the abstract and in the text, you refer to the appraisal tool as ROB 2 to avoid misunderstandings

Answer: Thank you for your insightful suggestion. We agree that referring to the appraisal tool as ROB 2 is more appropriate to avoid any confusion with other quality assessment tools. We will revise both the abstract and the main text of the manuscript to consistently refer to the tool as ROB 2, ensuring clarity for the readers. (Line 39,163)

3. Please mention the use of the GRADE approach in the abstract. In the text, please make sure that you use an appropriate original reference for GRADE and describe the instrument and the process in some more detail. For example, are you planning to use GRADEpro? Please mind that results of the GRADE assessments are “certainty (not grade) of the evidence”

Answer: Thank you for your helpful suggestion. We will revise the abstract to include a mention of the GRADE approach and its role in assessing the certainty of the evidence. In the main text, we will ensure that we use the appropriate original reference for GRADE and provide a more detailed description of the instrument and the process. Specifically, we plan to use GRADEpro to conduct the GRADE assessments, and we will clarify that the results of the GRADE evaluation refer to the certainty of the evidence. We will also make sure to use the correct terminology and methodology to avoid any confusion. (Line 39,40,168-174，445-449)

4. For screening of results, EndNote seems not sufficient, please state the use of an appropriate software as e.g. Covidence, Rayyan or EppiReviewer

Answer: Thank you for your suggestion. We agree that using specialized software for screening, such as Covidence, Rayyan, or EppiReviewer, would be more appropriate. We will revise the manuscript to reflect the use of EndNote and Rayyan for the screening process to improve the efficiency and accuracy of the review. (Line 119-123)

Reviewer #3:

General comments:

1. Does the manuscript provide a valid rationale for the proposed study, with clearly identified and justified research questions?

Reviewer #3: Yes

2. Is the protocol technically sound and planned in a manner that will lead to a meaningful outcome and allow testing the stated hypotheses?

Reviewer #3: Yes

3. Is the methodology feasible and described in sufficient detail to allow the work to be replicable?

Reviewer #3: Yes

4. Have the authors described where all data underlying the findings will be made available when the study is complete?

Reviewer #3: Yes

5. Is the manuscript presented in an intelligible fashion and written in standard English?

Reviewer #3: Yes

Answer: Thank you for your comments and recognition of our research.

Specific Comments:

1. Subgroup Analysis

Consider including a sensitivity analysis to assess the robustness of findings.

Answer: Thank you for your suggestion. We appreciate the importance of sensitivity analysis in evaluating the robustness of our findings. We would like to clarify that sensitivity analysis is already included in our methodology, specifically by changing the pooled model, conducting meta-regression, and excluding low-quality studies. To enhance clarity, we have now explicitly separated sensitivity analysis from subgroup analysis in the text and provided further details to emphasize its role in confirming the robustness of our conclusions. (Line 249-251, 260-270)

2. Discussion

More emphasis should be placed on the practical implications of using FICB over systemic analgesia, particularly in resource-limited settings.

Answer: Thank you for your valuable feedback. We agree that emphasizing the practical implications of using fascia iliaca compartment block (FICB) over systemic analgesia, especially in resource-limited settings, is crucial. We have revised the discussion section to further elaborate on how FICB can be a more effective and feasible option in such environments. Specifically, we have highlighted its advantages in reducing opioid consumption, minimizing systemic side effects, and potentially improving patient outcomes in settings where access to multimodal analgesia and advanced monitoring may be limited. We appreciate your insightful suggestion and believe this revision strengthens the clinical relevance of our study. (Line 311-321).

---

## [Editor Report · Decision Letter 3]

12 Feb 2025

Comparison of traditional systemic analgesic, Single shot or continuous Fascia Iliaca Compartment Block for pain management in patients with hip or Proximal Femoral Fractures: A protocol for systematic review and network meta-analysis

PONE-D-23-33546R3

Dear Dr. Tang,

We’re pleased to inform you that your manuscript has been judged scientifically suitable for publication and will be formally accepted for publication once it meets all outstanding technical requirements.

Kind regards,

Sascha Köpke

Academic Editor

PLOS ONE

---

## [Editor Report · Acceptance letter]

PONE-D-23-33546R3

PLOS ONE

Dear Dr. Tang,

I'm pleased to inform you that your manuscript has been deemed suitable for publication in PLOS ONE. Congratulations! Your manuscript is now being handed over to our production team.

Kind regards,

on behalf of

Professor Sascha Köpke

Academic Editor

PLOS ONE